# ✴ Polaris: Scaling Up Instruction-Guided Image Generation Towards Millions of Personalized Needs

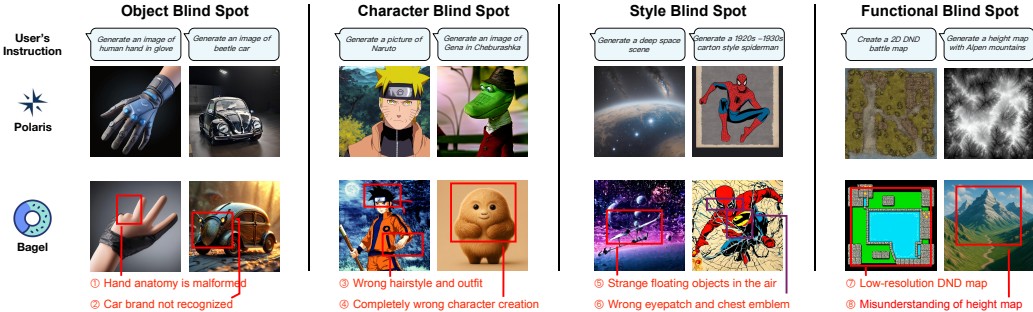

Figure 1: Although large models fine-tuned on trillions of tokens have enhanced their ability to understand diverse user instructions, such as style generation, they still exhibit significant blind spots. We introduce Polaris, a retrieval framework over the Stable Diffusion model zoo that automatically identifies and invokes the most relevant models, addressing these blind spots and enabling diverse, personalized image generation.

## Abstract

Users increasingly expect image generation models to quickly adapt to highly diverse and personalized requirements, such as producing images with distinctive styles or characteristics. Traditional approaches rely on fine-tuning, which is costly and difficult to scale. To cope with these limitations, the community has accumulated a growing library of fine-tuned modules and adapters, where each component targets specific generation needs and collectively serves as a foundation for handling new demands. This naturally raises a question: *instead of repeatedly training new models, can we systematically exploit this expanding ecosystem to better fulfill user instructions*? To this end, we present Polaris, an intelligent retrieval framework that automatically selects and integrates suitable models from the model zoo based on a user's instructions. The key insight is that harnessing such a massive and heterogeneous pool requires not only finding the most relevant modules among thousands of candidates, but also aligning them effectively for instruction-driven generation and editing. Polaris addresses this challenge by indexing over 6,500 checkpoints and 75,000 adapters, and retrieving the most relevant components given a user's input and instruction. In doing so, it delivers scalable, controllable, and well-aligned generation—without any additional training.

## 1 Introduction

The field of image generation has witnessed remarkable progress, particularly with the emergence of diffusion models that demonstrate strong capabilities in producing high-quality, diverse images (Song et al., 2020; Rombach et al., 2022; Podell et al., 2023; Yang et al., 2023). These models have been successfully applied to a wide range of practical downstream tasks, including super-resolution (Gao et al., 2023), image completion (Saharia et al., 2022), style transfer (Qi et al., 2024), image seg-

mentation (Tian et al., 2024), and image editing (Brooks et al., 2023a). Such advances have made generative models increasingly accessible to end users.

Users' needs are often highly diverse and deeply personalized with real-time feedback. For example, they may wish to generate images in distinctive artistic styles, replicate niche visual domains, or perform complex editing operations on existing content. However, a single base model frequently falls short, as it is typically pre-trained on a narrow set of common use cases with high computational cost. Even parameter-efficient training strategies such as LoRA (Hu et al., 2022) partially alleviate this issue, they remain costly, demand careful data preparation, struggle to generalize to the long tail of personalized demands, and fail to scale to the breadth of real-world requirements.

Building on this context, it is important to recognize that many users have already adopted more lightweight strategies in practice. Instead of training models from scratch, which is costly and time-consuming, they often start from a base model and leverage community-released checkpoints or adapters that capture distinctive styles, domains, or editing capabilities. Such components are openly shared through platforms including Civitai, PixAI, and Tensor.art, which collectively host tens of thousands of publicly available resources. This ecosystem substantially reduces the cost of customization and has enabled the rapid prototyping of diverse generative workflows. Motivated by these observations, we ask whether these community-contributed modules can be systematically harnessed to extend and promote the capabilities of our model. If feasible, this paradigm would provide a lightweight and scalable alternative to traditional personalization methods, while simultaneously raising important challenges in component selection and integration for downstream tasks such as instruction-driven generation and editing.

However, systematically exploiting this rich and heterogeneous ecosystem is far from trivial. Two core challenges arise. First, user queries for controllable image generation are often semantically complex and multimodal, requiring the system to jointly interpret open-ended natural language instructions and visual references. Second, as the pool of checkpoints and adapters grows to tens of thousands, retrieval must be efficient and scalable; naïve search quickly becomes impractical for interactive generation. To address these challenges, we introduce Polaris, a unified framework for instruction-driven model selection. Polaris translates user queries into effective model selection across style, object, and semantic dimensions by formulating the problem as a multimodal retrieval task. It extends beyond text-only input by integrating large language models (LLMs) as a zero-shot signal for improved instruction understanding, and further incorporates an efficient reranking strategy to support large-scale search without sacrificing responsiveness.

With these designs, Polaris systematically harnesses community-contributed checkpoints and adapters to deliver instruction-driven image generation without requiring any additional training. It can directly parse user instructions, recommend the most suitable models, and produce high-quality outputs. Compared with approaches that optimize solely for instructional input, Polaris achieves substantial gains by leveraging its large-scale model zoo. Moreover, as shown in Figure 1, relative to large pretrained models, Polaris complements their capabilities by improving performance on blind spots outside their training distribution, while also enjoying clear advantages in inference efficiency. Together, these results establish Polaris as a scalable and effective paradigm for controllable image generation and editing. Our contributions can be summarized as follows:

- We emphasize the need to address highly diverse, user-specific requirements while maintaining efficiency, and demonstrate that leveraging a large model zoo offers a practical and flexible solution.
- Our Polaris retrieves models using combined text and image queries and incorporates an efficient reranking strategy, enabling fast and scalable selection from tens of thousands of candidates.
- By harnessing the model zoo, Polaris delivers high-quality outputs that adapt to diverse user instructions with real-time performance, enabling scalable and customizable image generation.

## 2 RELATED WORKS

### 2.1 DIFFUSION-BASED TEXT-TO-IMAGE GENERATION.

Diffusion models generate images by progressively denoising random noise under the guidance of a learned score function, which enables high-quality and diverse synthesis. Stable Diffusion (Rombach et al., 2022) further introduced the latent diffusion framework, operating in a compressed latent space

to achieve an efficient tradeoff between computational cost and image fidelity. Since its initial release, multiple versions (v1, v2, SDXL (Podell et al., 2023), and the recent SD3/3.5 (Esser et al., 2024)) have been developed, continually improving resolution, fidelity, and controllability, making diffusion a dominant paradigm for text-to-image generation.

## 2.2 Model Adaptation for Personalized Generation

Model adaptation for personalization can be divided into checkpoint-based and adapter-based methods. The former, such as full fine-tuning (Ruiz et al., 2023b) and DreamBooth (Ruiz et al., 2023a), directly update model parameters to learn new concepts, producing new checkpoints but often at the cost of data efficiency and forgetting. The latter, including LoRA (Hu et al., 2022) and Hypernetworks (Ha et al., 2016), insert lightweight adapters that preserve the base weights and support modular, efficient customization. In general, checkpoint-level adaptation enables stronger style or domain shifts, while adapter-based methods are better suited for fine-grained or localized edits.

Early model zoos (Ramesh & Chaudhari, 2021; Falk et al., 2025) were constructed at the checkpoint level, collecting full model states for reuse and comparison. With the growing adoption of adapters such as LoRA, recent work has shifted toward adapter-level zoos (Huang et al., 2023; Zhao et al., 2024; Luo et al., 2024). However, these approaches leave several gaps: they do not establish a clear index linking checkpoints and their associated adapters, they only support retrieval at a single granularity (either checkpoints or adapters), and in the image generation domain (e.g., Stylus) adapter retrieval is restricted to a single modality without support for multimodal search.

## 3 Preliminaries

In this section, we first revisit the conventional instruction-guided image generation task and discuss representative approaches. We then introduce our extended formulation with personalized needs, which targets diverse and user-specific requirements and introduces new challenges.

### 3.1 Instruction-Guided Image Generation

Instruction-guided image generation aims to produce images that follow both a given text instruction $I$ and, optionally, a reference image $T$. A common paradigm for this task is based on Stable Diffusion (SD), where the generation process is controlled by a model configuration. Typically, such a configuration includes a checkpoint $c$, which defines the base generative capability, and a LoRA adapter $l$, which provides task- or style-specific adaptations corresponding to users' needs.

To better align generative models with user intent, existing methods mainly take two directions: enhancing diffusion models with instruction-following ability, and leveraging multimodal large language models (MLLMs) to unify reasoning with generation. For instance, InstructP2P (Brooks et al., 2023b) fine-tunes Stable Diffusion on synthetic instruction–image pairs, enabling edits guided directly by natural language.

More recent work extends LLMs or MLLMs to incorporate image synthesis within a unified framework (Zhang et al., 2025). These approaches either model text and images jointly in an autoregressive manner (e.g., Emu (Sun et al., 2023), LaVIT (Jin et al., 2023), Chameleon (Team, 2024)), or combine autoregressive reasoning with diffusion-based generation (e.g., Transfusion (Zhou et al., 2024), LMFusion (Shi et al., 2024)). Such unified designs represent a promising step toward more general-purpose foundation models for controllable image generation with common user intent.

### 3.2 Instruction-Guided Image Generation with Personalized Needs

While existing methods achieve good performance, directly fine-tuning models for each individual user is impractical in real-world scenarios. User requirements are inherently diverse, and a single monolithic model cannot adequately capture such variation with high efficiency. A more practical solution is to leverage the growing ecosystem of community-shared resources, including both base checkpoints and lightweight adapters for specific needs. We therefore extend the *Model Zoo* beyond conventional checkpoints to also include adapters, enabling compositional model configurations.

Formally, we define the model zoo $\mathcal{Z} = \mathcal{C} \times \mathcal{L}$, where $\mathcal{C} = \{c_1, \ldots, c_{N_c}\}$ denotes the set of base checkpoints and $\mathcal{L} = \{l_1, \ldots, l_{N_l}\}$ the set of LoRA adapters specialized for different styles, domains, or objects. In practice, we collect over 6,500 checkpoints and 75,000 adapters from Civitai [1], resulting in millions of possible model configurations for scalable and highly customizable image generation.

In this setting, our goal is to leverage the Model Zoo to adaptively match user requirements with suitable model configurations. Specifically, given a user query $q = (I, T)$, we aim to retrieve the most relevant checkpoint $c$ and adapter $l$ from $\mathcal{Z}$ and use them for generation or editing. The process involves three key steps: (1) obtaining a representation of the query that captures both textual and visual intent, (2) retrieving the checkpoint and LoRA adapter that best align with this intent, and (3) performing image generation or editing with the selected model configuration.

**Multi-Modal Instruction Embedding.** To represent the query, we define an embedding $z_q$ that serves as the input to model retrieval. Rather than fusing $I$ and $T$ in a fixed manner, we select the dominant modality based on the type of instruction. Formally, we define:

$$z_q = \begin{cases} f_{\text{text}}(T), & \text{if } \beta(q) = \text{text-dominant}, \\ f_{\text{img}}(I), & \text{if } \beta(q) = \text{image-dominant}, \end{cases}$$

where $f_{\text{text}}$ and $f_{\text{img}}$ are the text and image encoders, and $\beta(q)$ denotes a modality selection rule defined over the instruction semantics.

This selection reflects a key assumption in our setting: different types of instructions emphasize different input modalities. For example, style transfer or global transformations rely more on $T$, while object-centric edits guided by image content rely more on $I$.

**Retrieval Objective.** Each candidate checkpoint $c \in \mathcal{C}$ and adapter $l \in \mathcal{L}$ has a precomputed embedding $\phi_c$, $\phi_l$. The similarity between the query and each candidate is computed as:

$$s_c(q, c) = \langle z_q, \phi_c \rangle, \qquad s_l(q, l) = \langle z_q, \phi_l \rangle.$$

We select the most relevant model components by:

$$c^* = \arg\max_{c \in \mathcal{C}} s_c(q, c), \qquad l^* = \arg\max_{l \in \mathcal{L}} s_l(q, l).$$

The final image $x^*$ is synthesized by applying the selected model configuration $\Theta_{c^*, l^*}$ under the guidance of the instruction $T$:

$$x^* = \arg\max_{x} p_{\Theta_{c^*, l^*}}(x \mid f_{\text{text}}(T)).$$

Through the model zoo paradigm, diverse user needs can be flexibly satisfied by selecting appropriate checkpoints and adapters. Nevertheless, two core challenges remain: (1) how to construct multi-modal embeddings that faithfully capture user intent under heterogeneous instructions, and (2) how to perform retrieval efficiently over a large repository of model components.

## 4 METHOD

### 4.1 OVERVIEW OF POLARIS

We aim to enhance instruction-guided image generation within our model zoo, facing two key challenges. First, user queries are often semantically complex. We address this with an *Instruction Parser*, which uses a vision–language model (VLM) to decompose instructions, and a *Region Masker*, which generates localized modification masks. Second, the multimodal nature of inputs complicates checkpoint and adapter retrieval. We tackle this with a *Multimodal Retriever* across heterogeneous sources, further accelerated by an *LLM Tree Rank* strategy. An overview is shown in Figure 2.

---

[1] https://civitai.com

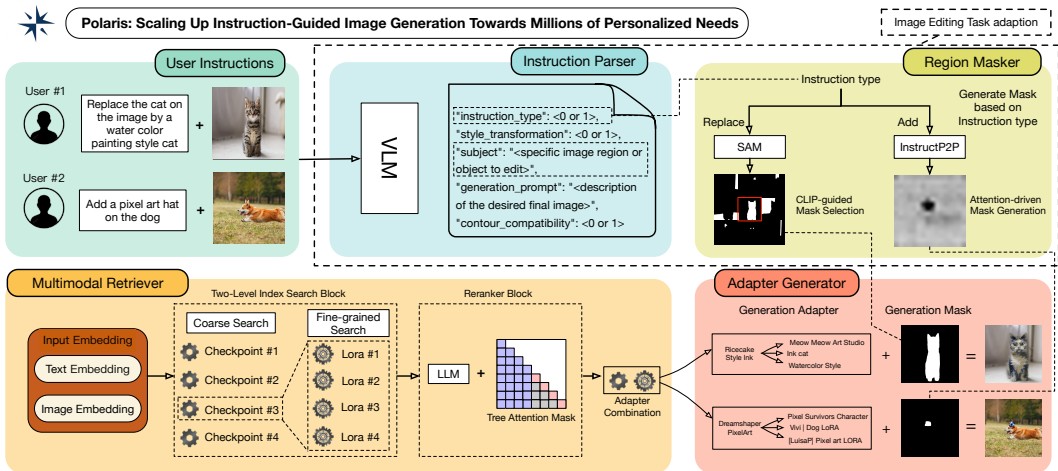

Figure 2: An overview of Polaris for instruction-guided model selection. The method is divided into two parts: (1) an Instruction Parser and Region Masker for adapting to image editing, and (2) a Multimodal Retriever for finding suitable checkpoints and LoRA adapters based on user requirements. The retrieved adapters and generated masks are combined to guide diffusion-based image generation.

## 4.2 INSTRUCTION PARSER

Given a user query $q = (I, T)$, where $I$ is an input reference image and $T$ is the textual instruction, the Instruction Parser is responsible for interpreting the user's intent and converting it into a structured representation that can be used for downstream retrieval and generation. To achieve this, we leverage a vision-language model (VLM) to parse $T$ in the context of $I$ when applicable. The parser decomposes the instruction into three main elements, which we formally define as a mapping:

$$\mathcal{P} : (I, T) \mapsto (t_1, t_2, t_3),$$

where $t_1$ denotes the *instruction subject*, i.e., the target object or region to be modified or generated; $t_2$ denotes the *instruction type*, specifying the operation (e.g., "modify", "replace", "stylize"); and $t_3$ denotes the *generation prompt* $T'$, a refined or reformulated version of $T$ designed to better align with the conditional diffusion model.

For example, given an instruction such as "Make the dog in this image look cartoonish", the parser identifies "dog" as the subject, "stylize" as the type, and generates a prompt like "A cartoon-style dog with exaggerated features." This structured understanding ensures that both the retrieval process and the final generation can be more precisely conditioned on the user's true intent. In our implementation, we leverage Qwen2.5-VL-7B (Bai et al., 2025) as the vision-language model to perform this instruction parsing. For a more detailed description of the prompts used by the instruction parser, please refer to the Appendix C.1.

## 4.3 REGION MASKER

The Region Masker grounds the instruction subject $t_1$, identified by the Instruction Parser, onto the user-provided image $I$. Its role is to localize the target region that should be modified while preserving unrelated areas.

**General Case ($t_2 = $ modify-type).** We first apply a pre-trained segmentation model to generate a set of candidate masks $\mathcal{M} = \{m_1, m_2, \ldots, m_{N_m}\}$. For each mask, we compute a relevance score using a value function $v(\cdot)$ that measures the semantic similarity between the mask region and the parsed subject. The optimal mask is then selected as:

$$m^* = \arg \max_{m \in \mathcal{M}} v(m, t_1).$$

The selected mask $m^*$ provides spatial constraints for the Adaptive Editor, ensuring that edits remain localized. In practice, we use the Segment Anything Model (SAM) (Kirillov et al., 2023) to propose

candidate regions, and compute relevance using a CLIP-based similarity function adapted to masked image inputs (Liang et al., 2023). Here, CLIP (Hessel et al., 2021) refers to a vision–language model trained to align images and text in a joint embedding space, making it well suited for evaluating mask–subject correspondence.

**Special Case ($t_2 = $ add-type).** For instructions involving object addition, SAM alone is insufficient. To better capture the intended region, we intersect the attention map from InstructP2P with the SAM-generated masks (Li et al., 2024), yielding a refined mask for the most relevant area to insert.

## 4.4 MULTIMODAL RETRIEVER

We construct two modality-specific indices offline: one for checkpoints $\mathcal{C}$ and one for adapters $\mathcal{L}$. Each item—whether a base checkpoint or a LoRA adapter—is embedded into a shared representation space that combines textual and visual information using a CLIP encoder. Formally, given a model $x$ (either a checkpoint $c$ or an adapter $l$), we compute its text embedding from the associated metadata—such as name, tags, release notes, or user-provided descriptions—and, if exemplar images $I_{x,1}, \ldots, I_{x,m}$ are available, we also compute its average image embedding:

$$\mathbf{z}_x^{\text{text}} = \text{CLIP}_{\text{text}}(\text{metadata}_x) \quad \text{and} \quad \mathbf{z}_x^{\text{img}} = \tfrac{1}{m} \sum_{i=1}^{m} \text{CLIP}_{\text{img}}(I_{x,i}).$$

Both components are concatenated to yield the final multimodal representation:

$$\mathbf{z}_x = [\mathbf{z}_x^{\text{text}} \, \| \, \mathbf{z}_x^{\text{img}}] \in \mathbb{R}^{2d}.$$

If no exemplar images are provided, we set $\mathbf{z}_x^{\text{img}} = \mathbf{0}$ to preserve dimensional consistency. The resulting embeddings are stored in the checkpoint index $\mathcal{C}$ and adapter index $\mathcal{L}$, enabling unified retrieval across modalities. To select with a user query $q$, we adopt a two-stage retrieval strategy:

**Level-1: Coarse checkpoint selection.** We first identify the most relevant base model by computing similarity scores $s_c(q, c)$ between the query and all entries in $\mathcal{C}$:

$$c^* = \arg \max_{c \in \mathcal{C}} \ s_c(q, c).$$

**Level-2: Fine-grained adapter selection.** Conditioned on the selected checkpoint $c^*$, we restrict the adapter search to a local neighborhood $\mathcal{L}_{c^*} \subseteq \mathcal{L}$ and identify the best-matched LoRA:

$$l^* = \arg \max_{l \in \mathcal{L}_{c^*}} \ s_l(q, l).$$

The final pair $(c^*, l^*)$ is then used to generate the output image conditioned on the user instruction $T$, and optionally a reference image $I$. To further improve adapter selection, we incorporate an LLM-based refinement module inspired by Stylus (Luo et al., 2024). The LLM decomposes user instructions into sub-tasks and allocates LoRA adapters, which provide more precise control over global style and object-level edits.

## 4.5 ADAPTIVE EDITOR

To enable flexible and precise image editing, our system integrates three distinct types of adapters: (1) a base checkpoint that governs the overall image style, (2) a style LoRA that modulates finer stylistic elements, and (3) an object-specific LoRA that controls the visual characteristics of specific entities (e.g., animal breeds or object categories).

The selection strategy for these adapters is conditioned on the nature of the user instruction. For instructions that focus on object replacement without explicit style control (e.g., "Replace the cat in the image with another breed"), we retrieve the overall style checkpoint and the fine-grained style LoRA from the input image to preserve visual consistency with the original scene, while extracting the object-specific LoRA based on the textual instruction. In contrast, for instructions that involve both object manipulation and explicit style transformation (e.g., "Replace the cat with a watercolor-style one"), we retrieve the overall style checkpoint and style LoRA from the textual prompt to reflect the desired stylistic shift, while still relying on the object-specific LoRA derived from the prompt to control entity-level appearance. Further details on how we distinguish between these instruction types and perform retrieval accordingly are provided in the appendix C.2.

### 4.6 Efficiency Optimizations in LLM-Based Reranking

In the *Multimodal Retriever* stage, we use LLM to refine retrieved candidates based on textual compatibility with user instructions and model metadata, similar to Stylus's reranking strategy. However, each LoRA adapter is associated with a textual description that must be provided to the LLM. As a result, when the candidate set is large, the prompt length can exceed 20000 tokens. This results in significant computational overhead due to the quadratic growth of LLM self-attention with sequence length.

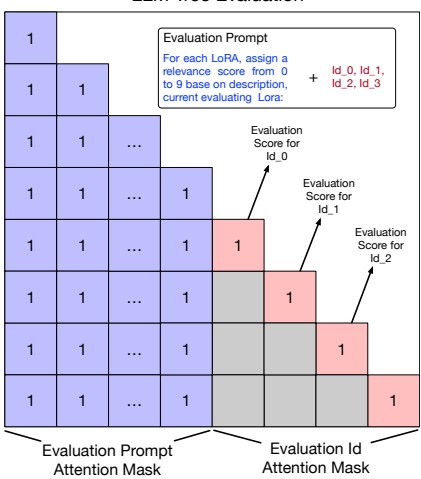

To address this, we propose a novel inference acceleration strategy called *Tree Reranking*, which balances retrieval quality and computational cost, making the reranking process more efficient and practical for deployment. The key idea is to modify the LLM's attention mask so that multiple branches (representing different adapter candidates) can be scored in a single forward pass. As illustrated in Figure 3, the reranker generates scores for multiple LoRAs simultaneously, then prunes low-scoring candidates and performs additional rounds of reranking on the remaining branches. By progressively narrowing the candidate pool, we drastically reduce the effective input length required at each stage. This tree-based speculative reranking effectively reduces redundant computation in attention mechanism, which would otherwise scale non-linearly with sequence length. In practice, our Tree Reranking approach yields a substantial speedup for the reranking module while maintaining high-quality adapter selection.

Figure 3: An overview of the tree rerank method. By modifying the attention mask of the LLM, we enable the model to evaluate multiple adapters simultaneously while generating only a single token per evaluation.

## 5 Experiment

**Experimental Setup.** In this work, we focus on instruction-guided image generation and propose to enhance performance through a model zoo. The zoo, consisting of model checkpoints and adapters, enables retrieval of suitable components according to user instructions. This design improves generation quality, especially in blind spots beyond the reach of conventional training. Further construction and implementation details of the model zoo are given in Appendix B.

**Evaluation Protocol.** To assess instruction-following ability, we evaluate on five tasks from GEDIT-BENCH (Liu et al., 2025): (i) background change, (ii) style change, (iii) material alteration, (iv) subject replacement, and (v) subject addition. These tasks cover a broad range of editing challenges, from appearance-level adjustments to semantic-level modifications. However, we observe that existing benchmarks often fail to capture the diversity of real user needs.

To bridge this gap, we construct a supplementary benchmark, called User-Bench, derived from community usage. In practice, users frequently upload the outputs of models to online communities after applying them to their own prompts, thereby providing both the input (the user prompt) and the corresponding output (the generated image). Leveraging this process, we curate a natural test set that reflects authentic user demands, and conduct a series of evaluations based on this dataset. The dataset construction process is detailed in Appendix D. We referred to GEDIT-BENCH and employed VLM as the evaluation tool. In contrast to their approach, we applied a more fine-grained evaluation dimension. The prompts used for the GPT-4o evaluation are provided in Appendix E.

### 5.1 Evaluation on Blind Spots in Image Generation

We evaluate our method on **blind spots**—cases where existing finetuned models systematically fail to generate reliable outputs, despite being trained on massive datasets. Such blind spots often arise from factors such as data imbalance or catastrophic forgetting. To address this challenge, our method leverages retrieval from a model zoo, avoiding additional finetuning while substantially improving

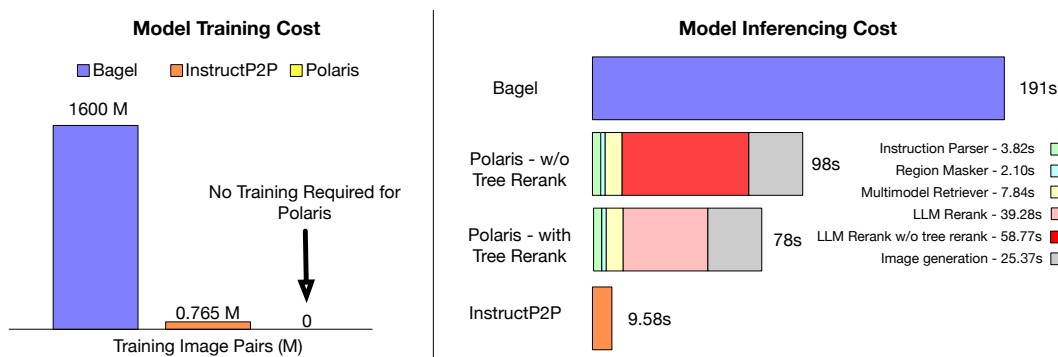

Figure 4: Comparison of training and inference efficiency across methods. Our approach, Polaris, requires no additional training and achieves inference time between the two baselines. Furthermore, by incorporating our proposed LLM Tree Rerank strategy, Polaris attains a 1.50× speedup in reranking, highlighting its practical efficiency advantage.

performance. We categorize blind spots into four types: object blind spots, character blind spots, style blind spots, and functional blind spots. As shown in Figure 1, our approach consistently outperforms the unified multi-modal baseline across all categories, producing more faithful and diverse results.

## 5.2 EVALUATION: QUANTITATIVE AND QUALITATIVE

Table 1: User-Bench results on two subcategories: Local style change and Style extraction. VQ (Visual Quality), EQ (Edit Quality), and their geometric mean (Overall) are reported. Polaris achieves significant gains on local style change compared to InstructP2P and attains competitive performance with unified multi-modal models, while obtaining the best results on style extraction.

|  | Style extraction | | | Local style change | | |
| --- | --- | --- | --- | --- | --- | --- |
|  | VQ | EQ | Overall | VQ | EQ | Overall |
| InstructP2P | 4.41 | 4.08 | 4.24 | 5.11 | 5.24 | 5.17 |
| BAGEL | 6.86 | 4.27 | 5.41 | 6.20 | 6.81 | 6.50 |
| Polaris | 7.33 | 5.50 | 6.35 | 5.84 | 6.19 | 6.01 |

Table 2: The experimental results on Gedit-Bench (Liu et al., 2025) demonstrate the effectiveness of our strategy. Compared with InstructP2P, which relies on finetuning to support instructional inputs, our approach leverages retrieval from a model zoo to achieve superior performance.

|  | Background change | Material alter | Style change | Subject add | Subject replace |
| --- | --- | --- | --- | --- | --- |
| InstructP2P | 3.70 | 3.39 | 4.60 | 3.18 | 3.80 |
| BAGEL | 7.06 | 6.40 | 6.13 | 8.06 | 6.71 |
| Polaris | 4.31 | 4.57 | 5.00 | 3.23 | 3.76 |

**Quantitative Evaluation.** On our User-Bench, we evaluate two tasks: Local Style Change and Style Extraction. Local Style Change, which involves editing within about 30 predefined styles, shows that our method outperforms InstructP2P and is comparable to Bagel, though its advantage is limited by the small and fixed style set. In contrast, Style Extraction treats each user request as a unique style, leading to a dramatic increase in style diversity and better reflecting real-world usage. As shown in Table 1, our method demonstrates clear superiority, particularly in rare or hard-to-describe styles, since our embedding-based style retrieval captures nuanced user-specific requirements that traditional text-prompt-driven models often miss.

On GEdit-Bench (Table 2), our framework delivers strong performance on common editing cases, achieving results on par with instruction-finetuned models such as InstructP2P. Notably, in scenarios involving substantial style transformations, our approach exhibits clear advantages, underscoring its effectiveness in handling complex edits. Although large unified multi-modal models achieve higher

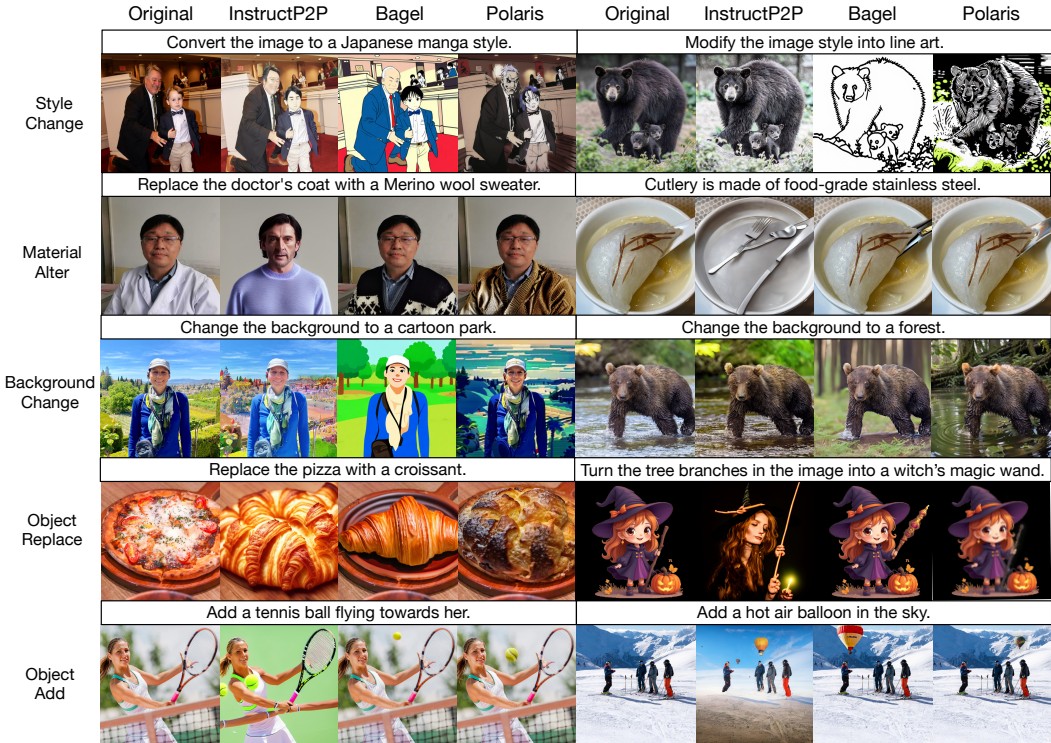

Figure 5: Qualitative results on different tasks of GEDIT-BENCH (Liu et al., 2025). Polaris achieves notably higher instruction-following success rates and more faithful outputs than InstructP2P, effectively handling style transformations that remain challenging even after fine-tuning. Moreover, the knowledge coverage of Polaris is comparable to Bagel, a trillion-token scale pretrained model.

overall scores, our framework—built upon conventional Stable Diffusion—comes remarkably close, highlighting the competitiveness of our lightweight and modular design.

**Qualitative Evaluation.** In Figure 5, we present qualitative comparisons o n GEdit-Bench. On standard editing tests, our approach exhibits much stronger instruction following than InstructP2P, producing edits that are both faithful and visually coherent. Remarkably, in terms of visual quality, our results are comparable to those of Bagel, a trillion-token scale pretrained model. Additional qualitative results on User-Bench are provided in Appendix A.

## 5.3 EFFICIENCY

We evaluate efficiency from two perspectives: training cost and inference latency, as shown in Figure 4. Our method is **completely training-free**, incurring zero training overhead. In contrast, Bagel and InstructP2P require ∼1.6M and ∼765K image pairs, respectively, making our approach far more practical and deployment-ready without costly retraining. For inference, our method runs at ∼78s per image, faster than Bagel (∼191s) though slower than InstructP2P (∼9.5s). Moreover, by incorporating our LLM-based tree reranking strategy, we further achieve a **1.50x speedup** in the reranking stage. Considering the elimination of training cost, our approach achieves a highly favorable balance between efficiency and applicability.

## 6 CONCLUSION

We presented Polaris, a retrieval-based framework that scales instruction-guided image generation by leveraging large pools of community-contributed models. By unifying multimodal retrieval with efficient adapter selection, Polaris improves alignment with user intent while remaining efficient and adaptable. This work demonstrates the potential of retrieval-driven approaches for achieving scalable and personalized generative systems.

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

## A  ADDITIONAL EXPERIMENT RESULTS

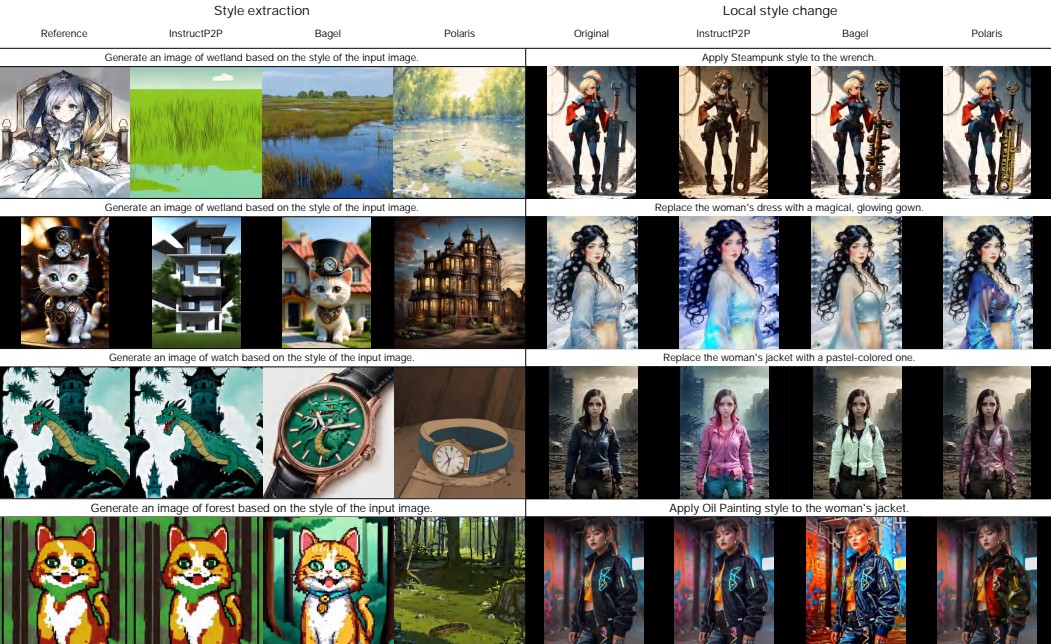

Figure 6: Experimental results on User-Bench. In the style extraction task, our method outperforms both baselines in terms of instruction understanding and generation quality. In the local style change task, our results surpass InstructP2P and are comparable to those of Bagel.

## B  IMPLEMENTATION DETAILS

We use Stable Diffusion 1.5 (SD 1.5) as the baseline generative model for all experiments, as it is the most widely adopted open-source backbone with extensive community-contributed adapters. We collect models from the Civitai platform, excluding those containing NSFW content. To build our retrieval system, we construct a model zoo comprising approximately 6,700 community-shared checkpoints, each representing a user-uploaded variant of SD 1.5, together with about 75,000 LoRA adapters (split following (Luo et al., 2024)). Among these adapters, roughly 9,000 are dedicated to style generation and 65,000 to object generation, each paired with embeddings that encode their style or task specifications. For reranking retrieved candidates, we employ Qwen2.5-14B as a large language model (LLM) reranker, which improves alignment between user instructions and the selected checkpoints or LoRAs. Both the base diffusion model and all variants in the database are implemented within the SD 1.5 framework, ensuring compatibility and controllability across the system.

## C  METHOD DETAILS

### C.1  INSTRUCTION PARSER

To interpret user inputs, we employ a vision–language model (VLM) guided by a tailored prompt. The parser analyzes the instruction along several dimensions: *instruction type* (categorizing the input task), *style transformation* (whether a style change is required), *subject* (the target entity of modification), *generation prompt* (translating instructions into standard prompts since SD 1.5 does not support direct instruction inputs), *contour compatibility* (whether large-scale shape changes are involved), and *foreground elements* (which foreground objects are to be edited). The full prompt design is provided in Table 3.

### C.2 ADAPTER EDITOR

During image generation, we employ three types of adapters to influence the final output: *style checkpoints*, *style LoRAs*, and *object LoRAs*. Depending on the instruction, different modalities are used for retrieval. For instructions that do not involve style transformation, we aim to preserve the original style of the edited image. In this case, a style checkpoint and a style LoRA are retrieved using image embeddings, while the object LoRA is retrieved using text embeddings. For instructions that explicitly involve style transformation, the original style is not preserved; instead, the style checkpoint, style LoRA, and object LoRA are all retrieved using text embeddings.

## D DATASET GENERATION

The evaluation dataset is constructed from user-uploaded images on Civitai, covering generations produced by diverse models. To ensure fair testing, we specifically select outputs generated by models with architectures different from SD 1.5. This structural difference keeps the test data disjoint from both the training data of the baseline model and the retrieval database, ensuring that neither the base model nor the retrieved checkpoints and LoRAs have been exposed to the evaluation samples.

We design two types of test cases: *style extraction* and *local style change*. For the *local style change* task, we employ `Qwen2.5-VL-7B` to automatically generate prompts conditioned on the downloaded images. The complete prompt design is summarized in Table 4. For the *style extraction* task, we first construct a subject pool and then randomly sample one subject. The resulting textual instruction is formulated as: "Generate an image of <subject> based on the style of the input image."

## E EVALUATION PROMPT

We use GPT-5o-mini to evaluate the final results on User-Bench. For the two tasks, *Local Style Change* and *Style Extraction*, we design different evaluation prompts to better capture task-specific objectives. The prompt used for assessing style transformation in image editing is summarized in Table 5.

## F LLM USAGE

Large language models (LLMs) were used only for minor language editing and polishing of the manuscript. They were not involved in the design of the research, development of methods, execution of experiments, analysis of results, or generation of scientific content. The authors take full responsibility for the final content of the paper.

**Given a user's image editing request in natural language, extract the following elements:**

**1. "instruction_type": Identify the editing intent.**

- 0 = Category change
• Condition: The target object's core category **changes to a different class**.
Decide **only** by category, **not** by verbs like "replace/change".
If A and B share the same high-level category, do **not** use 0.
• Examples: "change the cat to a dog", "replace a chair with a table"
• Note: If the target category stays the same but only style/appearance changes
(e.g., "replace the cat by a watercolor cat"), use 2 instead.
- 1 = Add new object
• Condition: A new object/element is introduced into the scene.
• Examples: "add a hat to the person", "put a bird in the sky", "add a tree next to the house"
- 2 = Local style/appearance transformation
• Condition: The object category remains the same, but its style/appearance changes.
• Examples: "make the cloth black and white", "turn the cat into a watercolor cat", "change the shoe into a red shoe"
• Special case: If the modification explicitly targets the "background" (e.g., "adjust the background to a forest", "make the background
blurry"), treat it as type 2, since "background" is a specific region rather than the whole image.
- 3 = Global style/appearance transformation
• Condition: A style/effect is applied to the entire image.
• Examples: "make the image black and white", "apply a watercolor filter to the whole picture"
- 4 = Style transfer from reference image
• Condition: The user provides a reference image and requests generating new content
(objects, scenes, characters, etc.) **in the style of that reference image**.
• Key difference: Unlike 0–3, this does **not** edit the original image; the reference serves only as a style source.
• Examples: "generate a cat in the style of the reference image", "create a cityscape with the style from the given artwork".

**2. "style_transformation": Binary flag (0 or 1).**
- Relevant for instruction_type = 0 or 1.
- 1 = The request explicitly specifies a style or artistic effect.
• Example: "turn the cat into a watercolor dog" $\rightarrow$ 1
• Example: "add a watercolor cat" $\rightarrow$ 1
- 0 = No style/effect is mentioned.
• Example: "change the cat to a dog" $\rightarrow$ 0
• Example: "add a cat" $\rightarrow$ 0
- For instruction_type = 2 or 3, this value is always 1 by definition.

**3. "subject"**
Must strictly follow this structure $\rightarrow$ "<TARGET_TYPE>+ <LOCAL_RANGE>".
- <TARGET_TYPE>: The core object name, no possessive forms.
• Correct: "cloth", "shoe"
• Incorrect: "girl's cloth", "man's shoe"
- <LOCAL_RANGE>: Spatial area where the target is located.
• Examples: "on the upper body", "in the foreground", "near the tree"

**4. "generation_prompt"**
Describe only the **final appearance** of the object/region.
- Do NOT mention the editing action (replace/change/remove).
- Do NOT add attributes beyond the request.
- Keep it concise, faithful, and specific.

**5. "contour_compatibility": Strict binary flag (0 or 1).**
- 0 = Contour differs significantly.
• Examples: "flag $\rightarrow$ hat", "book $\rightarrow$ vase"
- 1 = Contour is compatible.
• Examples: "apple $\rightarrow$ orange", "car $\rightarrow$ bus"

**6. "foreground_elements"**
A JSON array listing the distinct foreground objects in the image request.
- Each element should be a simple noun phrase (e.g., "dog", "tree", "man", "hat").
- Keep them concise, no extra adjectives unless explicitly requested.
- MUST NOT be empty. If uncertain, infer plausible main foreground objects from the request context.

**Respond strictly in the following JSON format:**
```
{
  "instruction_type":  <0|1|2|3|4>,
  "style_transformation":  <0|1>,
  "subject":  "<specific image region or object to edit>",
  "generation_prompt":  "<final desired description>",
  "contour_compatibility":  <0|1>,
  "foreground_elements":  ["<object1>", "<object2>", ...]
}
```

**User's request:** "{prompt}"

Table 3: Analyst prompt specification. The table outlines the schema and annotation rules used to extract structured representations, including instruction type, style transformation, subject, generation prompt, contour compatibility, and foreground elements.

**You are given an input image.**
The target visual style (S) for this edit is: "`{style}`". You must apply this style as instructed below. Please follow these instructions carefully:

**1. First, Identify and enumerate ALL main subjects in the image.**
A 'subject' refers to any physically distinct person, animal, object, background/scene, or any major visible part of a person or animal, such as hair (hairstyle), face, upper body clothing (shirt, jacket, dress), or lower body clothing (pants, skirt, dress). Do not include minor details like shoes, socks, glasses, or small accessories as separate subjects.
- For any person or animal, list major visible parts as separate subjects if they are visually distinct (for example: (1) person's hair, (2) person's face, (3) person's shirt, (4) person's pants, etc.).
- For any object that a person or animal is interacting with (e.g., a guitar being played, a book being held), also include it as a separate subject.
- Do NOT combine multiple items or persons as one subject. List each main subject and each major visible part separately.
- Example list:
(1) Woman's hair
(2) Woman's face
(3) Woman's dress
(4) Guitar (being played by the woman)
(5) Microphone
(6) The stage background

**2.Count the number of subjects you listed, and use this as 'num_subjects'.**

**3. If `num_subjects` > 1:**
- Randomly select **only one** subject to edit (for example, choose subject (4): Guitar).
- Replace the selected subject with a different object, entity, or new scene.
- Apply the style "`{style}`" ONLY to the replaced/new subject. Do NOT apply this style to the entire image. All other subjects and the rest of the image should keep their original style and appearance.
- Clearly state which subject (by number and description) was chosen in your edit_instruction.

**4. If `num_subjects` = 1:**
- Do NOT replace the subject. Only apply the style štyleŏ the subject (and its background if the subject is the background itself).

**5. Under no circumstances should you replace or modify more than one subject at a time. Do NOT apply the style globally.**

**6. Your response must be a JSON object using the following format:**
```
{
  "subjects":  [ <a list of identified subjects as strings> ],
  "num_subjects":  <int>,
  "edit_instruction":  "<one concise sentence describing the single subject replacement
(if any) and the style change, clearly stating the chosen subject by number and
description>",
  "result_prompt":  "<a detailed description of the final image after editing, focusing
on what is visually present in the image.  The description should not mention any editing
actions or changes.  Only describe what can be directly seen in the resulting image.>"
}
```

**Do NOT include markdown, code fences, or commentary — return only the JSON object.**

Table 4: Prompt template provided to the VLM during dataset construction. It specifies how the model should enumerate image subjects, select a single editing target, apply the designated visual style locally, and output a structured JSON object representing the generated image editing instruction.

**System / Judge Instruction**

You are an image evaluation model. The evaluation target is **"Image Editing with Style Transformation."**

**Important Instruction:** You **must** always return a result, even if it's not perfect. Ensure that you provide the requested evaluation for the image modification, including the scores and the reasons.

**Task Definition**

The user provides a source image (with the object/region to be edited) and a text description. The model must keep the overall structure of the source image while **modifying the specified object/region**, transforming it into a new form and/or applying a new style.

**Inputs**

- Source image: <SRC_IMAGE>
- Candidate image: <CAND_IMAGE>
- User text prompt: <USER_TEXT>

**Evaluation Focus**

- **Style integration**: Does the new style of the modified part appear consistent and well integrated with the whole image?
- **Structural consistency**: Are unmodified regions preserved without unnecessary changes or corruption?
- **Image quality**: Is the generated image visually clean, stable, and free of major flaws?
- **actual_modification**: To what extent have real, meaningful modifications been made to the image?

**Scoring Rubric (Scores 0–4)**

| Style integration | Structural consistency |
|---|---|
| 0: Style completely wrong | 0: Severe redraw/corruption |
| 1: Slightly aligned | 1: Most areas degraded |
| 2: Some correct, poor integration | 2: Majority preserved, issues |
| 3: Largely consistent | 3: Largely preserved |
| 4: Highly consistent | 4: Fully preserved |

| Image quality | Actual modification |
|---|---|
| 0: Severe artifacts | 0: No real modification |
| 1: Major flaws | 1: Minor tweaks only |
| 2: Acceptable | 2: Substantial change |
| 3: Good quality | 3: Significant changes |
| 4: Polished | 4: Major mods, intact structure |

**Output JSON Schema**

```
{
  "style_integration":  0-4,
  "structural_consistency":  0-4,
  "image_quality":  0-4,
  "actual_modification":  0-4,
  "reasons":  {
    "style":  "< <=40 words>",
    "structure":  "< <=40 words>",
    "quality":  "< <=40 words>",
    "modification":  "< <=40 words>",
  }
}
```

Table 5: Evaluation prompt specification. The prompt is fed to GPT-5o-mini to verify style-transformed image edits. It defines the evaluation focus, scoring rubric, and the structured JSON schema required for standardized reporting of results.

