# OpenReview forum: "Polaris: Scaling Up Instruction-Guided Image Generation Towards Millions of Personalized Needs"
_ICLR.cc/2026/Conference — ICLR 2026 Conference Withdrawn Submission_

### Official Review · Reviewer_2rkZ · 2025-10-25

**Soundness:** 2
**Presentation:** 3
**Contribution:** 2
**Rating:** 4
**Confidence:** 4

**Summary:**

The mitivation of this work: User's needs to personalize images is very diverse and interactive, the community has accumulated a growing library of editing modules, where each module taregts for one specific personalization need. So, is it possible to systematically utilize the exisiting library to fulfill user's instructions?
> In the reviewer's intuition only readingt the abstract, it's based on the "overlaps" between a "new" user's need and existing customization adapters.

In order to do this effciently and effectively, the author proposed Polaris. As shown in Figure 2, there is an Instruction Parser and Region Masker design to interpret the user's instruction and localize the image regions to edit.
> Will there be a "region" if the instruction is about global style or the entire environment?

After that, there'll be a Multimodal Retriever to search across a large model library and select the most relevant components based on both text and image embeddings. And finally the personalization is implemented by selected adapters.

The experimental results, no matter in numbers or visualizations, demonsrates that Polaris clearly beats InstructP2P, and beats BAGEL in "style extraction" (Table 1). It's worthwhile to note that compared with BAGEL and InstructP2P, Polaris is training-free (Figure 4). Also, it more efficient than BAGEL (Figure 4).

**Strengths:**

1. This paper studies a very interesting problem, that how to effectively use existing libraries for image personalziation. The reviewer agrees that as the community continues to develop, the neceessity of training a new model for a specific personalization need is decreasing.
2. The writing / drawing is clear and easy to follow. The appendix is rich.

**Weaknesses:**

1. In the paper, the author "emphasize the need to address highly diverse, user-specific requirements" (line 095). But in your method, you have a "Instruction Parser" that manually classified the "type" of the user's personalization need. In the reviewer's side, they're contrary with each other.
2. Another potential issue is the capability of the candidates, the author claimed they search within 6,500 checkpoints and 75,000 LoRA adapters. Even though the user's need is diverse, is it diverse enough to have 75,000 categorys? The reviewer don't think so. Intuitively, there should be many candidates targeting on the same task. Shall we better try to rank their capabilities before selection?
3. Also, intuitively, there should be some strong candidates that can handle multiple tasks (take BAGEL as an inappropriate example) in good quality. And the pipeline of Polaris seems to "assume" each canidate targets one specific function.

**Questions:**

Please check the weaknesses above.

---

### Official Review · Reviewer_KJ4P · 2025-10-31

**Soundness:** 2
**Presentation:** 2
**Contribution:** 1
**Rating:** 2
**Confidence:** 3

**Summary:**

The paper introduces Polaris, a retrieval-based framework for instruction-guided image generation that aims to overcome the limitations of traditional fine-tuning. Instead of training new models for every user request, Polaris leverages the existing ecosystem Stable-Diffusion checkpoints and LoRA adapters, automatically selecting the most relevant components to satisfy a user’s textual and visual instruction.

**Strengths:**

1. Retrieval-driven approach. Polaris reframes personalized generation as a model retrieval problem, unifying multimodal embedding, instruction parsing, and efficient reranking. This perspective is distinct from prior fine-tuning or adapter-composition works. The introduction of a tree-structured attention-masking scheme for LLM-based reranking is suitable for large-scale retrieval efficiency.
2.
3. Writing. Writing is generally clear and structured, with good diagrams illustrating system flow and blind-spot examples.

**Weaknesses:**

The core premise of Polaris which retrievals over thousands of community fine-tuned checkpoints can substitute for unified fine-tuning is viable because such a large pool of pretrained models already exists. However, this dependence raises serious questions about long-term scalability and efficiency. Once a new foundation model is released (e.g., SDXL → SD3, or a new visual backbone), all existing adapters and checkpoints would need to be retrained or re-aligned for compatibility. However, this inefficient approach is the only way to improve Polaris’s performance and extend its functionality.

In contrast, recent unified models such as Bagel, MetaQuery, and Show-o2 etc. show that understanding, generation, and editing can be jointly modeled within a single foundation model, enabling smoother function and performance scaling. Thus, Polaris arguably represents a return to a fragmented paradigm: one task, one model, which modern unified frameworks are beginning to surpass. The paper’s efficiency argument (e.g., Figure 4) does not account for the aggregate cost of pretraining the retrieved checkpoints and LoRA adapters, making the data-efficiency claim potentially misleading.

**Questions:**

1. Scalability and model upragde.  How does Polaris handle the migration problem when the base diffusion backbone evolves (e.g., SD 1.5 → SDXL → SD3)? Wouldn’t all retrieved checkpoints and adapters need to be retrained for compatibility? How is this more efficient than unified fine-tuning in that scenario?

2. Data Efficiency. In Figure 4, you claim that Polaris requires 0 training data and hence data-efficient. However, the retrieved checkpoints themselves were pretrained on large datasets. Could you clarify how total data usage, including that used to train all community checkpoints, is normalized against Bagel or other unified models?

3. Unified Paradigm Comparison. Given that recent work (e.g., Bagel, Show-o2) demonstrates joint training for editing, generation, and style transfer, why is retrieval-based composition a preferable direction? Have you considered combining your retrieval framework with a unified model backbone to mitigate redundancy and improve long-term scalability?

---

### Official Review · Reviewer_uv1p · 2025-11-01

**Soundness:** 3
**Presentation:** 3
**Contribution:** 1
**Rating:** 4
**Confidence:** 4

**Summary:**

The paper introduces Polaris, a training-free, retrieval-based framework designed to scale instruction-guided image generation to highly personalized user needs. The core contribution lies in harnessing an extensive, open-source model zoo encompassing over 6,500 checkpoints and 75,000 community-contributed adapters. Given a user's instruction and an input image, the method proposes to retrieve the most relevant model and subsequently generate images using the retrieved model. This model selection strategy integrates several key components: an Instruction Parser, a Region Masker, and a Multimodal Retriever.

**Strengths:**

1. The model selection strategy, based on leveraging the vast ecosystem of community-released models, is interesting.
2. The paper is well-written and esay to follow.

**Weaknesses:**

1. The proposed model selection strategy, while demonstrating compelling results, is engineering-oriented and heavily relies on the pre-existing capabilities of community-released models. I think it is not a foundamental advance in improving generative model's capacity to deal with diverse or novel user demands.
2. A major concern is the reliance on a massive, heterogeneous model zoo (over 6,500 checkpoints and 75,000 adapters). The paper should clarify the criteria and robustness measures used for model selection. Specifically, how is the quality assurance, safety, and lack of content artifacts of these models ensured, and how does potential selection bias affect generalizability?
3. The method requires a high storage demand to store a large number of checkpoints and adapters.
4. For style transferrig task which involves changing the global style of the image, whether the Region Masker is also used? It seems that the Region Masker only generates the local mask.

**Questions:**

See weaknesses

---

### Official Review · Reviewer_J2np · 2025-11-03

**Soundness:** 1
**Presentation:** 3
**Contribution:** 1
**Rating:** 2
**Confidence:** 4

**Summary:**

The paper presents Polaris, a retrieval-based framework that harnesses large collections of fine-tuned diffusion models and adapters to enable personalized and instruction-aligned image generation. Instead of training or fine-tuning new models for each request, Polaris retrieves and integrates the most relevant pre-existing modules from extensive repositories of community-contributed models. By combining multimodal retrieval with efficient adapter selection and alignment, Polaris achieves flexible, scalable, and training-free adaptation to diverse user instructions.

**Strengths:**

* The paper tackles an interesting problem on how to efficiently reuse the growing number of fine-tuned diffusion models and adapters without retraining.
* The writing is clear, making the paper easy to follow.

**Weaknesses:**

* The paper does not provide insights into which models Polaris tends to select for each task. For example, what are the most frequently retrieved models on GEditBench? Does the retrieval process reveal any bias toward particular model families or styles? Such an analysis would offer interpretability and evidence that Polaris is making meaningful retrieval decisions rather than arbitrary selections.
* The retrieval process appears to rely solely on textual or task relevance, which is insufficient when multiple models can perform the same task. How does Polaris determine which model yields the best actual performance among those candidates? Without incorporating performance-based or compatibility-based metrics, the retrieval may be suboptimal, potentially leading to inconsistent generation quality.
* Although the model zoo is large and diverse, the experiments cover only five editing tasks out of the eleven available in GEditBench. The authors should justify this selection: why were the other tasks omitted? A more comprehensive evaluation on the full GEditBench would strengthen the paper’s empirical validity and demonstrate generality across a wider range of edit types.
* The paper compares Polaris only against InstructP2P and Bagel, which represent limited baselines in today’s image editing landscape. More recent and competitive methods such as OmniGen, FLUX.1 Kontext (dev), and Step1X-Edit should be included to contextualize performance. Without such comparisons, it is difficult to assess whether Polaris provides genuine state-of-the-art improvements.
* There is no ablation study isolating the effect of different retrieval strategies or different subsets of the model zoo. This makes it unclear which component contributes most to the overall performance gains.

**Questions:**

* Could you provide the distribution of retrieved models per task (e.g., histogram of selected checkpoints on GEditBench)?
* Why were only 5 tasks chosen for evaluation instead of the full 11 tasks in GEditBench?
* How does Polaris handle conflicts or redundancy when multiple adapters target similar styles or objectives?

---

### Note · Authors · 2025-11-14

I have read and agree with the venue's withdrawal policy on behalf of myself and my co-authors.